# Overexpression of a Poplar RING-H2 Zinc Finger, *Ptxerico*, Confers Enhanced Drought Tolerance via Reduced Water Loss and Ion Leakage in *Populus*

**DOI:** 10.3390/ijms21249454

**Published:** 2020-12-11

**Authors:** Min-Ha Kim, Jin-Seong Cho, Eung-Jun Park, Hyoshin Lee, Young-Im Choi, Eun-Kyung Bae, Kyung-Hwan Han, Jae-Heung Ko

**Affiliations:** 1Department of Plant & Environmental New Resources, Kyung Hee University, Yongin 17104, Korea; minha123@khu.ac.kr (M.-H.K.); jinsung3932@gmail.com (J.-S.C.); 2Division of Forest Biotechnology, National Institute of Forest Science, Suwon 16631, Korea; pahkej@korea.kr (E.-J.P.); hyoshinlee@korea.kr (H.L.); yichoi99@korea.kr (Y.-I.C.); baeek@korea.kr (E.-K.B.); 3Department of Horticulture, Michigan State University, East Lansing, MI 48824, USA; hanky@msu.edu

**Keywords:** abscisic acid, Arabidopsis, drought tolerance, ion leakage, poplar, transpirational water loss, XERICO

## Abstract

Drought stress is one of the major environmental problems in the growth of crops and woody perennials, but it is getting worse due to the global climate crisis. XERICO, a RING (Really Interesting New Gene) zinc-finger E3 ubiquitin ligase, has been shown to be a positive regulator of drought tolerance in plants through the control of abscisic acid (ABA) homeostasis. We characterized a poplar (*Populus trichocarpa*) RING protein family and identified the closest homolog of *XERICO* called *PtXERICO*. Expression of *PtXERICO* is induced by both salt and drought stress, and by ABA treatment in poplars. Overexpression of *PtXERICO* in Arabidopsis confers salt and ABA hypersensitivity in young seedlings, and enhances drought tolerance by decreasing transpirational water loss. Consistently, transgenic hybrid poplars overexpressing *PtXERICO* demonstrate enhanced drought tolerance with reduced transpirational water loss and ion leakage. Subsequent upregulation of genes involved in the ABA homeostasis and drought response was confirmed in both transgenic Arabidopsis and poplars. Taken together, our results suggest that *PtXERICO* will serve as a focal point to improve drought tolerance of woody perennials.

## 1. Introduction

Drought stress is one of the major environmental problems in the growth of plants, including both crops and woody perennials. In corn alone, $15 to $20 billion is lost worldwide every year [1]. Due to the global climate crisis and the predicted increase of the global population, it is imperative to establish a strategy to make both crops and woody perennials more tolerant to a water-limited environment. Biomass from woody perennials accounts for more than 90% of the total biomass produced on earth, and about 25% of the annual anthropogenic CO_2_ emissions can be assimilated during woody biomass formation, which suggests that woody perennials serve as one of earth’s major long-term terrestrial carbon sinks [2,3,4,5]. *Populus* is a well-known genus for the production of an ideal woody biomass, because of their fast growth with a suitable wood quality for various applications [6].

The RING (Really Interesting New Gene) domain-containing proteins are known to have E3 ubiquitin ligase activity, which controls protein degradation through the ubiquitin-proteasome system in various species [7,8]. In plants, the most abundant E3 enzymes were RING-type E3 ubiquitin ligases [9] and a total of 469 predicted RING proteins have been reported in Arabidopsis [10,11]. Many E3 ubiquitin ligases regulate plant growth and development. For example, COP1, the first RING finger gene from Arabidopsis, has been identified to negatively regulate photomorphogenesis [12], and BIG BROTHER controls organ size by reducing cell proliferation [13].

In addition, some E3 ubiquitin ligases regulate abiotic stress response, including drought tolerance [9,14,15,16,17]. XERICO (Greek for ‘drought tolerant’) from Arabidopsis has been shown to confer drought tolerance when overexpressed in Arabidopsis [18] and rice [19] through the increase of endogenous ABA level. Interestingly, XERICO homologs of monocot plants were found to have similar function, although the similarity of amino acid sequence is low. For example, overexpression of *OsRHP1* (rice RING-H2 zinc finger protein 1; 36.5% identity to XERICO) improved drought and salt tolerance of rice plants by increasing ABA contents [20]. More recently, *ZmXERICO1* (XERICO homolog from *Zea mays*; 32.5% identity to XERICO) was found to function as an E3 ubiquitin ligase and its overexpression confers improved drought tolerance in both Arabidopsis and maize [16].

Here, we characterize the expression and function of PtXERICO (64.9% identity to XERICO) from *Populus trichocarpa*. Our analysis showed that expression of *PtXERICO* is induced by abiotic stresses (e.g., salt and drought stress) and ABA treatment. Indeed, overexpression of *PtXERICO* in a hybrid poplar confers enhanced drought tolerance by decreasing transpirational water loss and ion leakage. In addition, genes involved in the ABA homeostasis (*PtNCED3* and *PtCYP707A2*) were upregulated in the transgenic hybrid poplars, implying the increased ABA accumulation. Use of *PtXERICO* for the improvement of drought tolerance of woody perennials will be discussed.

## 2. Results

### 2.1. PtXERICO, a RING-H2 Zinc Finger from Populus trichocarpa, Is Orthologous to XERICO of Arabidopsis

Previously, we demonstrated that XERICO from Arabidopsis (i.e., AtXERICO) functions as a positive regulator of drought tolerance through ABA homeostasis (Ko et al., 2006). We found the closest homolog of AtXERICO from *P. trichocarpa* by BLAST search (https://blast.ncbi.nlm.nih.gov/Blast.cgi), and named it *PtXERICO* (Figure 1). PtXERICO showed 80% similarity (64.9% identity) to AtXERICO with the conserved N-terminal transmembrane (TM) domain and a RING-H2 zinc finger motif located at the C-terminus (Figure 1a). Phylogenetic analysis of the six XERICO homologs showed an evolutionary conservation and a clear divergence between dicot and monocot clades (Figure 1b). *PtXERICO* is expressed ubiquitously, but abundant in both stem and buds in *P. trichocarpa* (Figure 1c). In addition, *PtXERICO* expression was highly induced by both salt and drought stress (Figure 1d). To determine the hormonal regulation of *PtXERICO* expression, various hormones, such as, ABA, gibberellic acid (GA), salicylic acid (SA), and jasmonic acid (JA) were treated for 0.5 h and 5 h. Interestingly, only ABA treatment showed substantial increases of *PtXERICO* from 0.5 h to 5 h (Figure 1e). Taken together, our results suggest that *PtXERICO* is most likely orthologous to *AtXERICO*.

### 2.2. Overexpression of PtXERICO Results in a Growth Retardation and Confers Salt and ABA Hypersensitivity in Arabidopsis

To investigate the functional significance of *PtXERICO*, we generated transgenic Arabidopsis plants overexpressing *PtXERICO* (35S::PtXERICO). We found that 15 out of 35 overexpression transgenic lines showed phenotypes like 35S::AtXERICO (line SS8-3, reported in Ko et al., 2006), such as short hypocotyl, round-shaped rosette leaves and growth retardation (Figure 2b–d; Appendix A). For further phenotypic characterizations, we selected three T3 homozygous lines (#18-16, high-level; #22-1, medium-level, and #30-2, low-level expression) based on their expression levels of introduced *PtXERICO* gene (Figure 2a).

Under salt stress (100 mM NaCl), seven-day-old 35S::PtXERICO seedlings showed a hypersensitivity, which was similar to that of 35S::AtXERICO (Figure 3a,b). Root growth was significantly reduced in the presence of 100 mM NaCl (Appendix A). Thus, we further tested the effect of ABA, a plant stress hormone involved in salt and drought adaptation. The growth of 35S::PtXERICO plants (#18-16) was arrested immediately after germination compared with WT plants at sub-micromolar concentration of exogenous ABA (0.3 μM), consistently with 35S::AtXERICO plants (Figure 3c,d).

### 2.3. Overexpression of PtXERICO Confers Drought Tolerance in Arabidopsis

To evaluate the drought stress tolerance of 35S::PtXERICO plants, we discontinued watering of 30-day-old WT and 35S::PtXERICO plants growing on the soil in a pot. Afterwards, the plants were kept in a growth room maintained at low humidity. As shown in Figure 4a, a representative picture of each treatment after 11 and 14 days without watering, 35S::PtXERICO plants exhibit striking drought stress tolerance when compared with WT plants. To quantify the drought tolerance of 35S::PtXERICO plants, we calculated survival rate by considering the completely dried plants to be dead, which cannot be revived after re-watering (data not shown). After 14 days without watering, all the WT plants were dead, while over 70% of 35S::PtXERICO plants had survived (Figure 4b).

Previously, the 35S::AtXERICO plants showed significant enhancement of drought tolerance by decreasing water loss via transpiration [18], we estimated the transpirational water loss by measuring the fresh weights (FW) of detached leaves over different times. The leaves from WT plants lost about 27% of their FW in 1 h, while leaves from 35S::PtXERICO plants had a significantly reduced water loss (approximately 18%), which is comparable to 35S::AtXERICO (~14%) (Figure 4c). After 5 h of the incubation, WT plants retained only 25% of water in the leaves, while 35S::PtXERICO plants retained 45%, which is 1.8-fold higher water retention than that of WT (Figure 4c).

### 2.4. Upregulation of PtXERICO Modulates the Expression of ABA Biosynthesis and ABA-Responsive Genes

To validate the drought stress tolerance of the 35S::PtXERICO plants (Figure 4), we examined the transcriptional regulation of *PtXERICO* and genes related to drought and ABA metabolism. Expectedly, upregulation of *PtXERICO* was shown in 35S::PtXERICO plants (#18-16 and #22-1) regardless of drought stress, like as *AtXERICO* in 35S::AtXERICO plants (Figure 5). Expression of endogenous *AtXERICO* is clearly upregulated by drought stress in WT plants, which was confirmed by employing a 3′-UTR primer of *AtXERICO* gene in the RT-PCR (Figure 5, *AtXERICO_UTR*). Interestingly, in the 35S::PtXERICO (#18-16) plants under drought stress conditions, the endogenous *AtXERICO* expression is not upregulated significantly, suggesting the orthologous function of PtXERICO to AtXERICO. Accordingly, *AtACS11* (*1-AMINOCYCLOPROPANE-1-CARBOXYLATE SYNTHASE 11,* At4g08040) was highly expressed in both the 35S::PtXERICO and 35S::AtXERICO plants, regardless of drought stress (Figure 5). *AtACS11* was found as the most highly upregulated gene in the whole transcriptome profiling data of 35S::AtXERICO compared to WT [18].

*NCED3* (*NINE-CIS-EPOXYCAROTENOID DIOXYGENASE3*; At3g14440) encodes a key enzyme in ABA biosynthesis [21], while *CYP707A2* (*ABA 8*′-*HYDROXYLASE*; At2g29090) is a key enzyme in the oxidative catabolism of ABA [22,23]. Transcripts of both *AtNCED3* and *AtCYP707A2* were clearly increased in the drought stress of 35S::PtXERICO plants, suggesting the increased accumulation of endogenous ABA in the 35S::PtXERICO plants like as 35S::AtXERICO plants (Figure 5). Clear upregulation of *AtRD29A* in the drought treatment (Figure 5), a well-known drought and ABA-inducible gene (At5g52310 [24]), further validated the overall drought stress treated experiments of Figure 4.

### 2.5. Transgenic Poplar Trees Overexpressing PtXERICO Exhibit Reduced Transpirational Water Loss

The significance of PtXERICO function was further characterized by generating transgenic poplar trees overexpressing *PtXERICO* (Figure 6). Hybrid poplars (*Populus alba* × *P. glandulosa*) were used in both wild type (clone BH) and transgenic plant experiments, respectively (see materials and methods). For comparative phenotypic analysis, we chose three lines of 35S::PtXERICO poplar trees (#21, high-level; #24, medium-level, and #3, low-level expression) based on their expression level of introduced *PtXERICO* gene (Figure 6a,b). Although no apparent phenotypic changes were observed in 35S::PtXERICO poplar trees compared to BH (Figure 6a), expression of genes involved in ABA homeostasis (i.e., *PtNCED3* and *PtCYP707A2*) were greatly upregulated, proportionally to the expression level of the introduced *PtXERICO* (Figure 6b). *PtNCED3* (Potri.011G112400.1) and *PtCYP707A2* (Potri.001G242600.1) were selected as the closest homologs of Arabidopsis *NCED3* (At3g14440) and *CYP707A2* (At2g29090), respectively. In addition, we tested the expression of the *PtACS11* (Potri.002G113900.1) gene in our 35S::PtXERICO poplar trees as an *ACS11* (At4g08040). Consistently, *PtACS11* was highly upregulated in the 35S::PtXERICO poplar tree (line #21) (Figure 6b).

Drought tolerance of 35S::PtXERICO poplar trees was also examined by employing the transpirational water loss as described above. The leaves from BH trees lost about 56% of their FW in 3 h, while leaves from 35S::PtXERICO poplar tree (line #21) had a significantly reduced water loss (approximately 39%) (Figure 6c). Furthermore, after 7 h of the incubation, BH trees retained only 22% of water in the leaves, while 35S::PtXERICO poplar tree (line #21) had 42%, which is 1.9-fold higher water retention than that of BH (Figure 6c).

### 2.6. Overexpression of PtXERICO Confers Enhanced Drought Tolerance in a Hybrid Poplar

To validate the drought stress tolerance of 35S::PtXERICO poplar trees, we discontinued watering of two-month-old BH (control) and 35S::PtXERICO poplar trees growing on soil. The 35S::PtXERICO poplar trees showed a dramatic drought stress tolerance when compared with BH plants. The line #21 of 35S::PtXERICO poplar trees showed no sign of wilting after 15 days of drought treatment (Figure 7a,b). Accordingly, expressions of *PtXERICO* and *PtNCED3* are increased significantly in both BH and 35S::PtXERICO poplar trees after drought treatment (Figure 7c). Very interestingly, expression of *PtNCED3* was drastically increased in BH trees by the drought stress while the expression of *PtNCED3* was much less increased in the 35S::PtXERICO poplar trees, especially in line #21 (Figure 7c). This result suggests that the 35S::PtXERICO poplar trees have a higher level of endogenous ABA.

Under environmental stresses, such as drought and salinity, plant cell membranes are damaged and lose their integrity. Thus, electrolyte leakage is increased [25]. However, under drought stress, 35S::PtXERICO poplar tree (#21) exhibited almost no changes of electrolyte leakage in the leaves, while BH showed a dramatic increase in electrolyte leakage from approximately 10% to 61% (Figure 7d). Thus, these results suggest that overexpression of *PtXERICO* reduces membrane damage caused by drought stress.

## 3. Discussion

Previously, overexpression of *XERICO*, an Arabidopsis RING-H2 zinc finger gene, was reported to confer drought tolerance through increased ABA biosynthesis in both Arabidopsis and rice [18,19]. Ko et al. [18] suggested that XERICO may function as an E3 ubiquitin ligase, because of its interaction with UBC8, an E2 ubiquitin conjugating enzyme. *OsRHP1* from rice and *ZmXERICOs* from maize showed enhanced drought tolerance with an increase of endogenous ABA level when overexpressed in both monocot and dicot plants [16,20]. These results demonstrate that they are orthologous to XERICO. Furthermore, ZmXERICO1 was found to function as an E3 ubiquitin ligase and affect the stability of ABA 8′-hydroxylase, which degrades ABA [16].

In this study, we found that *PtXERICO*, the closest homolog of *XERICO* from poplar (*P. trichocarpa*), shared a high sequence similarity with the conserved TM and RING-H2 zinc finger motif (Figure 1a). Phylogenetic analysis suggests the evolutionary conservation and divergence of different RING-H2 zinc finger proteins from either monocot or dicot plants (Figure 1b). Thus, PtXERICO was grouped together with AtXERICO in a dicot clade. Consistently with XERICO and its orthologues, *PtXERICO* expression was also induced by drought and salt stress (Figure 1d), and yielded hypersensitivity to salt and ABA treatment when overexpressed in Arabidopsis (Figure 3). However, we found that *PtXERICO* was upregulated by ABA treatment, which had not been reported previously in AtXERICO (Figure 1e). It is known that genes induced in response to exogenous ABA treatment are involved in the ABA-dependent stress response pathways [26,27]. Therefore, *PtXERICO* may be involved in the ABA-dependent stress response pathway. Recently, PeCHYR1, a ubiquitin E3 ligase from *Populus euphratica*, was reported to be upregulated by ABA treatment and enhances drought tolerance when overexpressed [28].

To further characterize the function of *PtXERICO*, we produced both transgenic Arabidopsis and hybrid poplar plants overexpressing *PtXERICO* and performed comparative phenotypic analysis by selecting three lines each having different expression level of introduced *PtXERICO* (Figure 2, Figure 3, Figure 4, Figure 5, Figure 6 and Figure 7). They exhibited enhanced water retention capacity, which was estimated by the transpirational water loss of leaves (Figure 4c and Figure 6c), consistently with the previous report [18]. The degree of resistance to transpirational water loss in both transgenic Arabidopsis and hybrid poplar plants is proportional to the introduced *PtXERICO* transcript level, further confirming the molecular function of PtXERICO (Figure 4 and Figure 6). Although the method we used (i.e., detached leaves) is generally acceptable, but has clear limitations. Thus, measurement of ‘on planta’ transpiration (e.g., gas exchange measurements, capillary flow porometry, sap flow measurements, and so on) will be included in the follow-up study.

Interestingly, unlike 35S::PtXERICO Arabidopsis plants, none of the 14 lines of 35S::PtXERICO poplar trees had significant growth retardations (data not shown). This may be due to the different sensitivity of hybrid poplar to ABA compared to Arabidopsis. Indeed, no visible phenotypical differences were reported between transgenic corn overexpressing ZmXerico1 and control [16].

The ability of cell membranes to control the ion flux in plants is used as a quantitative measurement for cell integrity under the various stress condition [25]. Previously, the electrolyte leakage of the sensitive maize cultivar increased from 11 to 54%, but the tolerant cultivar had much less electrolyte leakage [29]. Additionally, higher electrolyte leakage in drought stressed maize (*Zea mays* L.) plants was found than in plants grown under well-watered conditions [30]. The conductivity (i.e., electrolyte leakage) of BH poplars (i.e., control) increased by up to 61% after 15 day of drought treatment, indicating the severe damage of the membrane permeability (Figure 7d). This intracellular ion leakage caused plant senescence with wilting (Figure 7a,b). However, the transgenic poplars showed much less (line #3) or no detectable membrane damage (line #21) after drought stress (Figure 7d). 

Furthermore, our transgenic plants showed substantial upregulation of genes involved in ABA homeostasis (Figure 5, Figure 6 and Figure 7). Overexpression of *NCED3* gene enhanced cellular ABA level in various plant species, such as Arabidopsis, tomato, tobacco, and bentgrass [21,31,32,33]. Consistent with the 35S::PtXERICO Arabidopsis plants, both poplar *NCED3* and *CYP707A2* genes were upregulated significantly in the 35S::PtXERICO poplar trees (Figure 5, Figure 6 and Figure 7). These results suggest that the upregulation of *PtXERICO* regulates the expression of the ABA-biosynthetic gene, as well as the catabolic gene. Taken together, overexpression of *PtXERICO* probably upregulates endogenous ABA level in plants. Quantification of ABA contents will be necessary in our follow-up study.

In summary, we provide experimental evidence (e.g., analyses of gene expression and phenotypic characterizations of transgenic plants) that PtXERICO, a RING-H2 zinc finger from *P. trichocarpa*, is orthologous to Arabidopsis XERICO and overexpression of *PtXERICO* confers an enhanced drought stress tolerance in plants, most likely through the increase of cellular ABA level. Since *Populus* is known to their fast growth with a suitable wood quality, *PtXERICO* will serve as a focal point to improve woody biomass production by conferring drought tolerance.

## 4. Materials and Methods

### 4.1. Plant Materials and Growth Conditions

*Arabidopsis thaliana*, ecotype Columbia (Col-0, from Arabidopsis Biological Resource Center (https://abrc.osu.edu/)) and hybrid poplars (*Populus alba* × *P. glandulosa*, clone BH, from National Institute of Forest Science (https://nifos.forest.go.kr)) were used in both wild type and transgenic plant experiments, respectively. Plants were grown on soil or on half-strength MS medium (Murashige and Skoog, Sigma-Aldrich, MO, USA) with 1% (*w*/*v*) sucrose in a growth room (23 ± 2 °C, 14 h light (150 μmol·m^−2^·s^−1^)). Soil (Bio topsoil #1, Nongwoo Bio, Suwon, Korea) was a mixture of coco peat 47.2%, peat moss 35%, zeolite 7%, vermiculite 10%, dolomite 0.6%, wetting agent 0.006%, fertilizer 0.194%.

### 4.2. Vector Construction and Production of Transgenic Plants

The full-length cDNA encoding *PtXERICO* (Potri.014G170400.1) was amplified by polymerase chain reaction (PCR) from cDNA of poplar (*P. trichocarpa*). Subsequently, the amplified cDNA was inserted downstream of the 35S promoter in the pK2GW7 vector [34] using the Gateway cloning system (Invitrogen, Carlsbad, CA, USA) to produce 35S::PtXERICO construct. The vector construct was then introduced into *Agrobacterium tumefaciens* strain C58, which was used to transform both Arabidopsis and poplar by the floral-dip method [35] and leaf disk transformation-regeneration method [36,37], respectively. Primers used for amplification are listed in Appendix A.

### 4.3. RNA Extraction and RT-PCR

Total RNAs of Arabidopsis were extracted using the TRIZOL reagent (Life Technologies, Carlsbad, CA, USA). In addition, for RNA extraction of hybrid poplars, the cetyltrimethylammonium bromide (CTAB) method was used because of the high amounts of polysaccharides and polyphenols in poplars, as described previously [5]. One microgram of total RNA (260/280, ~2.0) was reverse-transcribed using Superscript III reverse transcriptase (Invitrogen, Carlsbad, CA, USA) in 20 μL reactions. Subsequent RT-PCR was performed with 1 μL of the two-fold diluted reaction product as a template with PCR program as, 1 cycle of 95 °C for 5 min, 28 cycles of 95 °C for 30 s, 60 °C for 30 s, and 72 °C for 1 min and 1 cycle of 72 °C for 5 min. RT-qPCR was performed using the AriaMx Real-Time PCR (G8830A) (Agilent, TX, USA) with Brilliant III Ultra-Fast SYBR^®^ Green QPCR Master Mix (Agilent, TX, USA). The reaction program was as follows: denaturation at 95 °C for 3 min, followed by 40 cycles of denaturation at 95 °C for 15 s and annealing at 60 °C for 30 s. The *PtActin2* gene (Potri.019G010400) was used as the internal quantitative control, and relative expression level was calculated by the 2-ΔΔCt method [38]. All primer sequences were designed using Primer3 software (http://fokker.wi.mit.edu). Sequences are provided in Appendix A.

### 4.4. Drought, Salt and ABA Treatments on Hybrid Poplar Seedlings

Four-week-grown hybrid poplars (clone BH) in test tubes (height: 15 cm, diameter: 2.2 cm) were used in this experiment. For drought treatment, we used a shock-like dehydration stress as described in [39]. Seedlings of hybrid poplars taken out from test tube were left on a clean-bench and sampled at 0, 3, and 8 h. For salt stress treatment, seedlings of hybrid poplars were moved to liquid MS media containing 150 mM NaCl and sampled at 0, 0.5, 1, 3, and 6 h. For hormone treatment, seedlings of hybrid poplars were moved to liquid MS media containing 50 μM of each hormone (ABA, GA, SA, and JA) and sampled at 0, 0.5, and 5 h. All samples (whole seedlings) were snap-frozen immediately in liquid nitrogen for RNA extraction.

### 4.5. Salt and ABA Treatment on Arabidopsis Seedlings

For salt stress treatment, Arabidopsis seedlings (WT, 35S::PtXERICO, and 35S::AtXERICO) were grown vertically in 1/2 MS-agar media without or with 100 mM NaCl for 7 days. For ABA treatments, Arabidopsis seedlings (WT, 35S::PtXERICO, and 35S::AtXERICO) were grown horizontally in 1/2 MS-agar media with 0, 0.1, 0.3 and 0.5 μM ABA for 7 days. Plant growth was quantified by measuring the fresh weight of 7-day-old seedlings. All experiments were performed in triplicate and repeated at least three times.

### 4.6. Drought Stress Treatments on Soil-Grown Arabidopsis and Hybrid Poplars

Progressive drought stress was treated as described in [40]. For Arabidopsis plants, WT and 35S::PtXERICO plants were grown on soil in a pot under the normal watering condition for 30 days and then discontinued watering. In the case of hybrid poplar, two-month-old normal irrigated soil-grown 35S::PtXERICO poplar trees and BH were discontinued watering. Afterwards, the plants were kept in a growth room maintained at low humidity.

### 4.7. Survival Rate Measurements

Survival rate (%) was measured at 0, 11, 12, 13 and 14 days after drought treatment by counting dead Arabidopsis plants. Completely dried plants of WT, 35S::PtXERICO, and 35S::AtXERICO that are not revived after re-watering were considered as dead plants.

### 4.8. Water-Loss Analysis

We used the method of detached leaves exposed to dehydration on a clean-bench, as described in [41]. In Arabidopsis, three fully expanded leaves from three WT and 35S::XERICO plants that were 30-day-old grown on soil were detached and left on a clean-bench. In the case of hybrid poplar, four-week-old soil-acclimated BH and 35S::XERICO poplar trees were used. The leaves were weighed at the indicated times to determine the rate of water loss.

### 4.9. Electrolyte Leakage Measurement

Electrolyte leakage (EL) was measured as described by [29], with slight modifications. Leaf discs (8 mm diameter) from 5~6th leaves of both BH and 35S::PtXERICO poplar trees were washed with deionized water, and then placed in tubes with 20 mL of deionized water and incubated for 15 h at 25 °C with gentle rocking (40 rpm). Subsequently, the electrical conductivity of the solution (L1) was measured by using conductivity meter (CON6 portable conductivity meter, Oakton, IL, USA). Samples were then autoclaved at 110 °C for 10 min and the final conductivity (L2) was measured after equilibration at 25 °C. The EL was defined as follows: EL (%) = (L1/L2) × 100.

### 4.10. Statistical Analysis

All experiments were performed in triplicate and repeated at least three times. The number of used plants is indicated in each result. Statistical analysis and graph generations were performed by using SigmaPlot v12.0 (Systat Software, Inc., Chicago, IL, USA). In addition, the significances of differences were calculated using Student’s *t*-test, represented by * (*p* < 0.05), ** (*p* < 0.01), and *** (*p* < 0.001). Error bars in graphs indicate standard error of mean.

## Figures and Tables

**Figure 1 ijms-21-09454-f001:**
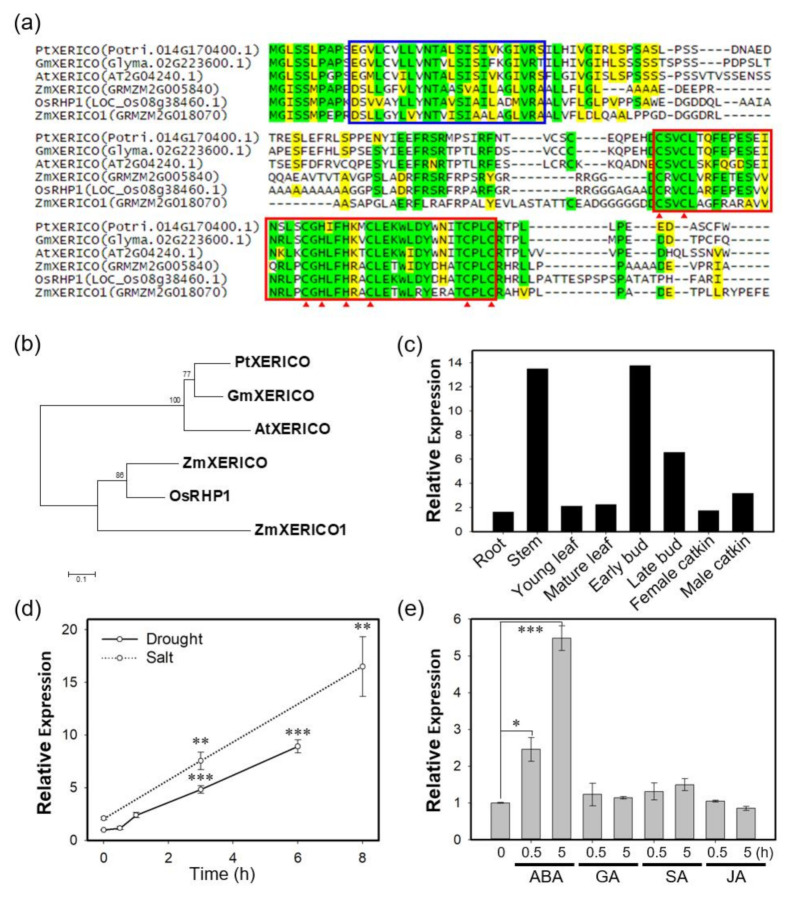
PtXERICO, a RING-H2 zinc finger from *Populus trichocarpa*, is orthologous to XERICO of Arabidopsis. (**a**) Amino acid sequence alignment of PtXERICO and related proteins from Arabidopsis (AtXERICO), soybean (GmXERICO), maize (ZmXERICO, ZmXERICO1) and rice (OsRHP1). Complete amino acid sequences were aligned using ClustalW. Putative transmembrane domains identified using TMHMM2.0 are indicated by blue box. RING-H2 zinc finger domains are identified by a red box and position of C and H by red arrowheads. Identical amino acids were shown in green while yellow colors indicate similar amino acids. (**b**) Phylogenetic analysis of proteins from (**a**). The rooted phylogenetic tree was constructed using a Neighbor-Joining method (1000 bootstrap replicates) and Jones-Taylor-Thornton (JTT) model in MEGA 7.0 (Kumar et al. 2016). (**c**) Tissue preferential expression of *PtXERICO* in *P. trichocarpa*. Plot was reconstructed from the public data from Phytozome (https://phytozome.jgi.doe.gov/pz/portal.html#!gene?search=1&detail=1&method=3252&searchText=transcriptid:27033190). (**d**,**e**) Expression of *PtXERICO* under the drought and salt stress (**d**) and hormone treatment (**e**). Four-week-old hybrid poplars (*Populus alba* × *P. glandulosa*, clone BH) grown in test tube were used (see, Methods). Student’s *t*-test, * (*p* < 0.05), ** (*p* < 0.01), and *** (*p* < 0.001).

**Figure 2 ijms-21-09454-f002:**
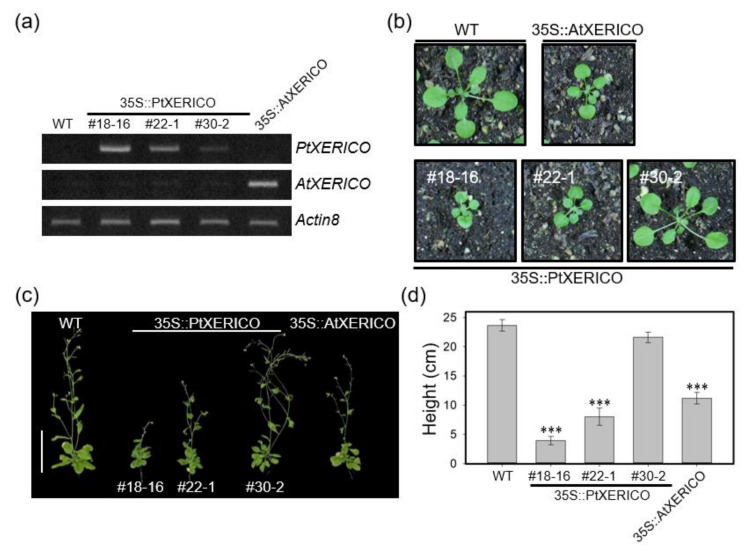
Overexpression of *PtXERICO* in Arabidopsis results in growth retardation. (**a**) Expression of *PtXERICO* transcripts in transgenic Arabidopsis plants (35S::PtXERICO). Transgenic Arabidopsis overexpressing XERICO of Arabidopsis (35S::AtXERICO, line SS8-3; Ko et al., 2006) was used as a positive control. Semi-quantitative RT-PCR (26 cycles) was performed using cDNA templates generated from leaf total RNA. The *Actin8* (At1g49240) gene was used as a loading control. (**b**,**c**) Growth retardation in both 35S::PtXERICO and 35S::AtXERICO plants. Twenty-day-old soil-grown plants (**b**) and fifty-day-old adult plants (**c**). Scale bar, 10 cm. (**d**) Quantification of plant growth. Heights of plants from (**c**) were measured. Error bars indicate standard error (*n* = 10). Student’s *t*-test, *** (*p* < 0.001).

**Figure 3 ijms-21-09454-f003:**
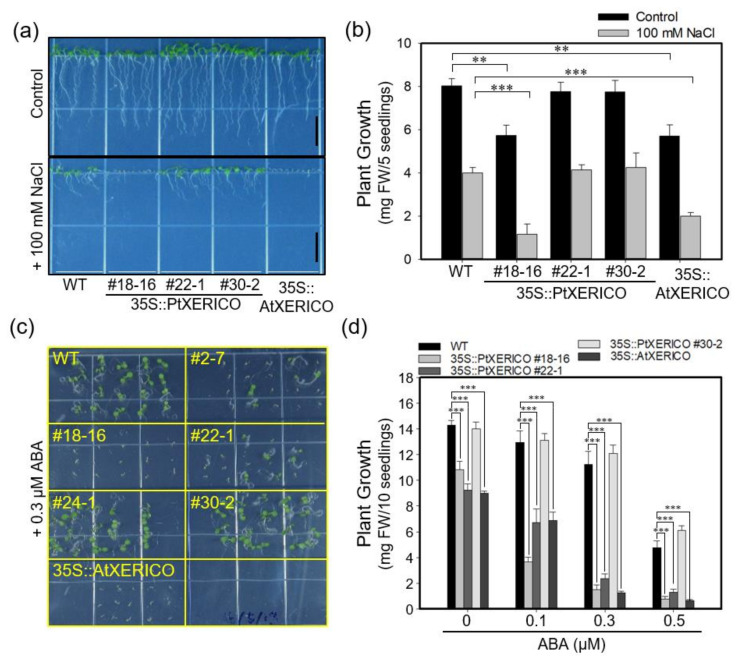
Overexpression of *PtXERICO* confers salt and ABA hypersensitivity in Arabidopsis. (**a**,**b**) Hypersensitive growth retardation of 35S::PtXERICO and 35S::AtXERICO plants to salt stress. Seven-day-old seedlings grown on MS media, vertically, without (control) or with salt (+100 mM NaCl) (**a**). Scale bar, 1 cm. In addition, quantification of the seedling growth shown in (**a**,**b**). Fresh weights were measured at 7 days after sowing. (**c**,**d**) Hypersensitive growth retardation of 35S::PtXERICO and 35S::AtXERICO plants to ABA treatments. Seven-day-old seedlings grown on MS media, horizontally, with 0.3 μM of ABA (**c**). In addition, quantification of seedling growth by measuring fresh weights at 7 days after sowing (**d**). Error bars represent the standard error of three independent experiments. Student’s *t*-test, ** (*p* < 0.01), and *** (*p* < 0.001).

**Figure 4 ijms-21-09454-f004:**
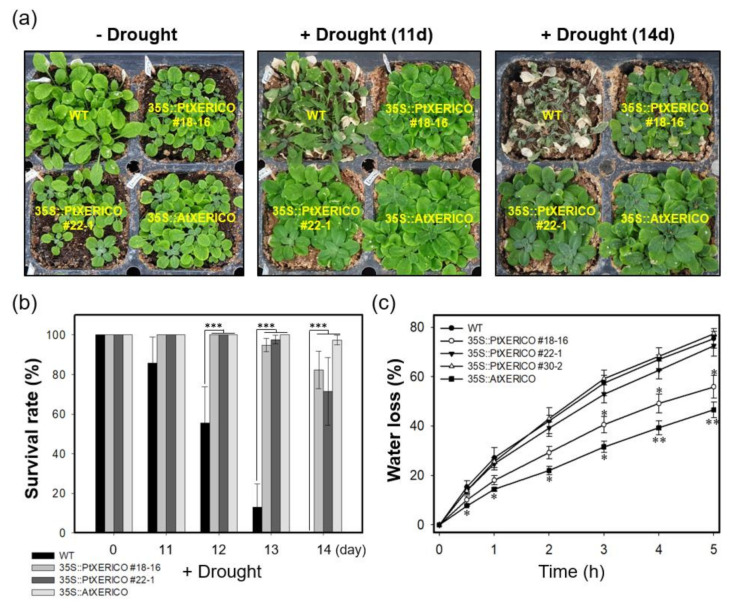
Overexpression of *PtXERICO* confers drought tolerance in Arabidopsis. (**a**) Drought stress treatment. Watering was discontinued in the 30-day-old WT, 35S::PtXERICO and 35S::AtXERICO plants growing on the soil in a pot. Representative pictures were shown 11 and 14 days without watering. (**b**) Survival rate measurement. Completely dried plants that are not revived after re-watering were counted as dead plants at 0, 11, 12, 13 and 14 days after drought treatment. (**c**) Transpirational water loss. Fully expanded leaves from 30-day-old WT, 35S::PtXERICO and 35S::AtXERICO plants were detached and left on a bench. The leaves were weighed at 0 to 5 h to determine the rate of water loss. Error bars represent the standard error of three independent experiments. Student’s *t*-test, * (*p* < 0.05), ** (*p* < 0.01), and *** (*p* < 0.001).

**Figure 5 ijms-21-09454-f005:**
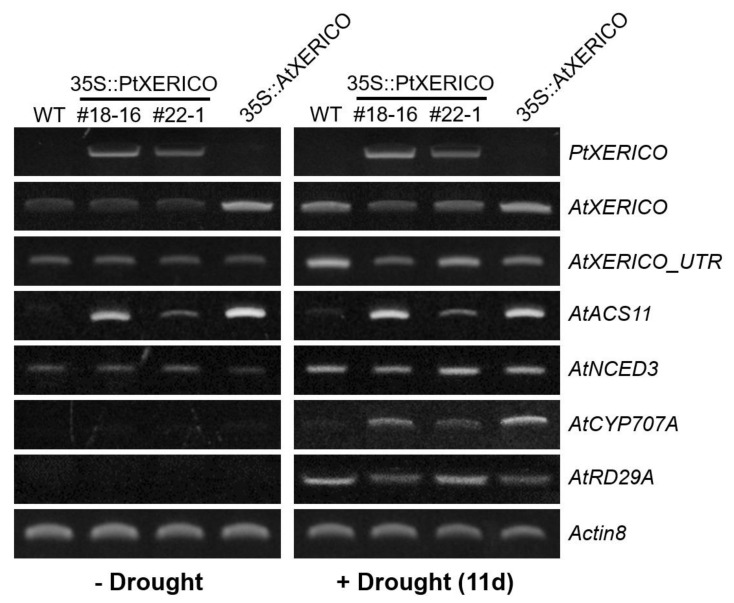
Upregulation of *PtXERICO* modulates the expression of ABA biosynthesis and ABA-responsive genes. A semi-quantitative RT-PCR was performed by using cDNAs obtained from ‘−Drought’ and ‘+Drought 11d’ Arabidopsis leaves of Figure 4a. *AtXERICO_UTR* indicates the endogenous *AtXERICO* transcripts by using a 3-UTR primer as a reverse primer. *AtACS11* (At4g08040); *AtNCED3* (At3g14440); *AtCYP707A2* (At2g29090); *AtRD29A* (At5g52310). *Actin8* (At1g49240) was used as a loading control. Primer sequences used in this study are listed in the Appendix A.

**Figure 6 ijms-21-09454-f006:**
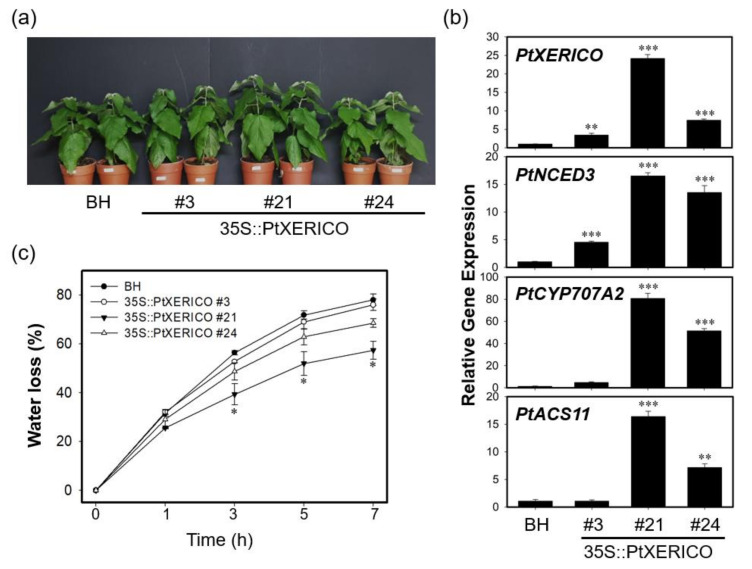
Transgenic poplar trees overexpressing *PtXERICO* exhibit reduced transpirational water loss. (**a**) Three-month-old soil-grown hybrid poplar (BH) and 35S::PtXERICO transgenic poplars. (**b**) Expression of *PtXERICO*, *PtNCED3* (Potri.011G112400.1), *PtCYP7072A* (Potri.001G242600.1) and *PtACS11* (Potri.002G113900.1). cDNA templates were generated from leaf total RNA of each 35S::PtXERICO transgenic poplars and BH. *PtActin2* (Potri.019G010400) was used as a quantitative control for the real-time quantitative PCR (RT-qPCR). (**c**) Measurement of transpirational water loss. Detached leaves of four-week-old soil-acclimated BH and 35S::PtXERICO plants were used. The leaves were weighed at 0 to 5 h to determine the rate of water loss. Error bars represent the standard error of three independent experiments. Student’s *t*-test, * (*p* < 0.05), ** (*p* < 0.01), and *** (*p* < 0.001).

**Figure 7 ijms-21-09454-f007:**
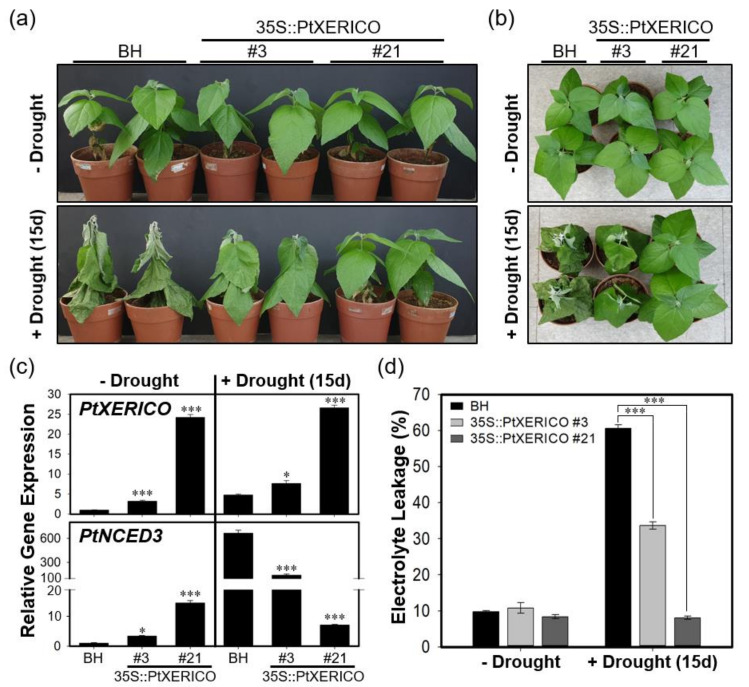
Overexpression of *PtXERICO* confers enhanced drought tolerance in a hybrid poplar. (**a**) Two-month-old soil-grown hybrid poplar (BH, control) and 35S::PtXERICO transgenic poplars without drought stress (upper panel, ‘−Drought’) and after 15 days of discontinuing watering (lower panel, ‘+Drought (15d)’). (**b**) Top view of (**a**). (**c**) Expression of *PtXERICO* and *PtNCED3* with or without drought stress treatment. cDNA templates were generated from leaf total RNA of each 35S::PtXERICO transgenic poplars and BH. *PtActin2* was used as a quantitative control for RT-qPCR. (**d**) Electrolyte leakage measurement. Leaves (5th~6th) of both BH and 35S::PtXERICO poplar trees were used to measure the electrical conductivity. Error bars represent the standard error of three independent experiments. Student’s *t*-test, * (*p* < 0.05), and *** (*p* < 0.001).

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
