# Peer review of "Overexpression of a Poplar RING-H2 Zinc Finger, Ptxerico, Confers Enhanced Drought Tolerance via Reduced Water Loss and Ion Leakage in Populus"

_ijms, 2020, doi:10.3390/ijms21249454_

Round 1

Reviewer 1 Report

The manuscript titled “Overexpression of a poplar RING-H2 zinc finger, PtXERICO, confers enhanced drought tolerance via reduced transpirational water loss and ion leakage in Populus” studying the function of RING-H2 zinc finger gene; XERICO in poplar using transgenic and transcriptional approaches. The authors attempted to identify the gene in poplar using the Arabidopsis gene as a query and investigated the hormonal regulation by studying gene expressions under hormone treatment as well as the role of the gene under stress. The topic is highly relevant from both scientific and practical points of view. However, there are main problems in the manuscript.
• The aim was not clearly mentioned in terms of the significance of the study and the whole manuscript is plagued with many sentences that are not clear and contain errors in structure and punctuation.
• The significance test for the results was not performed or indicated in the figures or the text. As well as not enough experimental details are provided.
• The authors have proposed that the tested gene confer drought tolerance through increasing ABA biosynthesis. Such proposed theory should be confirmed by analysis of endogenous ABA content in samples.
• The authors stated that overexpression of PtXERICO confers enhanced drought tolerance via reduced transpirational water loss however they did not measure the loss of water through transpiration in whole plants.
I recommend doing extensive revision for the manuscript. Please take the following notes in your consideration when revise your manuscript:
Materials and Methods: the section of methods miss some experimental details and the arrangement of sections is not good enough. Remember that your methods should be written in a clear fully described way to let others can do or repeat your work otherwise understand the flow of methodology in your work.
Survival rate measurement was not mentioned in the materials and methods section. I noted that the authors described the measurement in the results section, but it is not the right place for such information. Please indicate this accordingly in the M&M section.
4.3. RNA extraction and RT-PCR: the authors should clarify why they used two different methods for extracting RNA (using TRIZOL reagent and CTAB method).
Too many details are missed as information about RNA integrity, PCR program as the number of cycles, Tm and PCR efficiency the standard curve series used in the methodology.

Results: please indicate in figure 1d,1e the relative expression values from which tissue? Leaves tissue or stem or whole plant?
Please do a significance test for all results and show it on the figures as well as indicate the P-value in the figure caption.
Figures caption should contain all identification information that fully describe the figures. However, all words “indicated times” were repeated too many times in figure captions. Please list the time intervals instead.
Overall, the manuscript needs to rewritten and reorganized to address poor sentence structure and grammar; more details need to be provided in the text and figure legends to improve the quality of the manuscript. The discussion section needs to be improved.

Reviewer 2 Report

The manuscript IJMS-1018453 deals with very interesting and important topic of new gene identification in response to drought and dehydration in transgenic Arabidopsis and poplar. In general, the manuscript is very good, written very clearly, with the logical description and interesting results, which are well presented. However, moderate but very important revision is required listed below including missed sub-section for Statistical treatment with probability assessments and some important information regarding qPCR and terminology have to be added. Other listed minor corrections and notes can be addressed easily and no need responses from Authors to the Reviewer. Therefore, I would suggest accepting the manuscript conditionally with a subject to some revision and adding of a few more items mentioned below.

Major comments/suggestions:

(1) Sub-section 4.7 for Statistical treatment was missed. Please insert it with full description of the used statistical methods and software, as well as a number of used plants, biological and technical replicates in each experiment. Authors indicated most of the information in the Figure legends, where error bars means SE and it has to be also included in the sub-section 4.7. Additionally, Authors MUST think and select suitable method to assess statistical differences in each presented Figure with graphs at least between WT and presented transgenic lines. Authors can choose asterisks, letters or another favourite method to distinguish the probability level for the statistical conclusion. Error bars are good but not good enough for this purpose. Each Figure has to contain such probability level identifications, described in Figure legends as well as in sub-section 4.7.

(2) The sub-section 4.3. is very short where some details have to included: RNA integrity and quality control, DNase treatment during cDNA library construction, the composition and volume of qPCR mix including volume and concentration of qPCR primers. Please add the information that 1 ul of non-diluted synthesised cDNA was used as a template for qPCR because most of researchers are used several ul of diluted cDNA for such experiments. Please add brief description of qPCR program with the number of cycles, Tm and duration. It would be good to include information about qPCR efficiency and the standard curve series but this is not compulsory. Please add in the brackets that the PtActin2 gene was used as ‘the internal quantitative control (reference gene)’, indicating for Readers that these terms as synonyms.

(3) L331-342. There is an important terminology and methodology issue. Authors used three types of experiments: (A) Slow or progressive drought of whole plants in pots with soil (for example see, Ann. Bot., 2006, 97, 133-140); (B) Shock-like dehydration, where whole plants were taken from soil or media and exposed to dehydration on bench (for example, Funct. Integr. Genomics., 2009, 9, 377-396); and (C) Detached leaves exposed to dehydration on bench (for example, J. Plant Physiol., 2010, 167, 103-109). None of the indicated papers are belong to me and, therefore, this is not a ‘promotion’ of papers but this is my comment regarding the issue.

Authors described their results in experiments with type (A) in the sub-sections 2.3-2.6 (L130-246). However, none of the conditions was described for plant growth and slow/progressive drought treatment in the M&M section. Authors MUST present a new sub-section with full description of these experiments. Experiments with type (B) were described in sub-section 4.5 (L336-338) with indicated reference to Figure 1d. However, as I mentioned above, the treatment of pulled and exposed plants on bench is not drought but rather ‘Shock-like dehydration’, which is very differ from slow drought of plants in pots with soil. Please insert your comment in M&M section to clarify the dehydration procedure. Experiments with dehydration of detached leaves with type (C) are perfectly correct.

Minor corrections/notes:

(4) L35. It is better to use ‘more tolerant to a water-limited environment’, ‘drought tolerant’, or ‘tolerance to salinity’ because ‘resistant’ and ‘resistance’ are more suitable terms for ‘disease resistance’.

(5) L49. Please insert a full name in the first instance for the abbreviation ‘XERICO’.

(6) L78-79. The following Author’s statement is too strong: “Taken together, our results suggest that PtXERICO is orthologous to AtXERICO”. I am not against it but I suggest that Authors have to make more careful statement. Orthology between genes in different species can be found based on their phylogenetic analysis while expression profiles in response to abiotic stresses and hormonal treatments can confirm functional similarities between studied genes. Please make your modification of the sentence.

(7) L105-112. I can see a conflicting situation between plant images in Figure 2c and graphs with average data plus/minus SE for 10 plants in Figure 2d. The transgenic plant with 35S::AtXERICO genetic construct (last one in the Figure 2c) has a plant height just a little below of those in WT, and the height of this transgenic plant can be estimated as at least 20 cm based on the provided scale bar. However, results presented in Figure 2d are very different. It would be good if Authors can insert one sentence in Results or in another part with their brief explanation for Readers.

(8) L118, L127 and L341. Could Authors please check and make your corrections with consistent information about the concentration of applied hormones, for example ABA: (L118): 0.3 uM (microM); (L127): 0.3 mM (milliM); and (L341):50 uM (microM). Which information is correct, and which concentration was really used?

(9) L164 and L173. Please do not make a break in the middle of sentence. This is hard to follow the meaning.

(10) L187-188. The abbreviation ‘BH’ used twice nearby is confusing. Please use it once only in the second place as follows: ‘…(Populus alba x P. grandulosa) were used in both wild type (clone BH) and transgenic plant experiments…’

(11) L191-193. The sentence about transgenic corn with published reference is more suitable for Discussion rather than Results section.

(12) L206 and 209, Legend of Figure 6. Please be consistent and correct a spelling removing one letter ‘r’ in the trangene name, as it was written in L211: ‘35S::PtXERICO’ but not ‘35S::PtrXERICO’.

(13) L248 and L258-259. The name of the studied gene was ‘RING-H2 zinc finger’ as indicated in the title and L257 of the manuscript. Therefore, the name ‘RING-H2 finger’ gene or protein is inconsistent and incorrect without the word ‘zinc’.

(14) L305-311. Please add a few words where seed stock of Arabidopsis and poplar plant material were received from? If Authors generated poplar clones theirselves, please indicate it clearly.

(15) L337. Please add more information what kind and size of ‘test tubes; were used for growing of BH plants?

(16) L342. Please do not use ‘laboratory jargon’ with abbreviation ‘LN2’ but regular phrase ‘liquid nitrogen’ instead.

(17) Supplementary Table S1. Please add expected sizes of each amplified product in an additional column.

Round 2

Reviewer 1 Report

The manuscript has been extensively edited and the authors responded to all  comments of the reviewer. I see the manuscript now is ready for publication after deleting the word "transpirational" from the title of the manuscript to be "Overexpression of a poplar RING-H2 zinc finger, PtXERICO, confers enhanced drought tolerance via reduced water loss and ion leakage in Populus"

Author Response

Reviewer Comment 1:

The manuscript has been extensively edited and the authors responded to all  comments of the reviewer. I see the manuscript now is ready for publication after deleting the word "transpirational" from the title of the manuscript to be "Overexpression of a poplar RING-H2 zinc finger, PtXERICO, confers enhanced drought tolerance via reduced water loss and ion leakage in Populus"

Authors' Response1:

We really appreciate all of your efforts and precious comments on our manuscript (ijms-1018453). We modified the title as you suggested the following: Overexpression of a poplar RING-H2 zinc finger, PtXERICO, confers enhanced drought tolerance via reduced water loss and ion leakage in Populus.